# Si/SiGe QuBus for single electron information-processing devices with memory and micron-scale connectivity function

Ran Xue [1], Max Beer [1], Inga Seidler [1], Simon Humpohl[1,2], Jhih-Sian Tu[3], Stefan Trellenkamp[3], Tom Struck[1,2], Hendrik Bluhm [1,2] & Lars R. Schreiber [1,2] ✉

The connectivity within single carrier information-processing devices requires transport and storage of single charge quanta. Single electrons have been adiabatically transported while confined to a moving quantum dot in short, all-electrical Si/SiGe shuttle device, called quantum bus (QuBus). Here we show a QuBus spanning a length of 10 μm and operated by only six simply-tunable voltage pulses. We introduce a characterization method, called shuttle-tomography, to benchmark the potential imperfections and local shuttle-fidelity of the QuBus. The fidelity of the single-electron shuttle across the full device and back (a total distance of 19 μm) is (99.7 ± 0.3) %. Using the QuBus, we position and detect up to 34 electrons and initialize a register of 34 quantum dots with arbitrarily chosen patterns of zero and single-electrons. The simple operation signals, compatibility with industry fabrication and low spin-environment-interaction in $^{28}$Si/SiGe, promises long-range spin-conserving transport of spin qubits for quantum connectivity in quantum computing architectures.

Controlling local charge densities in a semiconductor by metallic gate-electrodes sets the foundation of modern nanoelectronics. The miniaturisation of gate-electrodes reveals quantum mechanical effects and entertains the development of nanoelectronics devices operating with single charge quanta. Discrete charge states of quantum dots (QDs) are stored to process digital information[1] and the spin of individual electrons is used to encode quantum bits for quantum computing in semiconductors[2,3]. The exchange of charge quanta between functional blocks such as charge–photon interfaces[4–6], quantum registers[7], spin manipulation zones[8], single charge detectors[9] and current standard devices[10] would lead to quantum devices with new functionalities.

For conventional electronics, wires transport currents or voltages over extended distances. In quantum technology, wires cannot transport individual charges, as disorder limits their localisation length

hardly exceeding 100 nm. In micron-sized quantum structures, the charging energy becomes impractically small for utilising charge states with definite electron number. A proposed device named quantum bus (QuBus)[11] solves this fundamental difficulty and might provide the key for the required[12,13] scale-up of quantum computing architectures[14–17].

Single electron[18,19] and spin-conserving electron[20] shuttling have previously been demonstrated employing surface acoustic waves in piezoelectric semiconductors. Shuttling in non-polar materials such as silicon, which is highly attractive for quantum computing with electron-spins[21–24], becomes more involved, as the electron transport requires a series of top gates. However, this additional complexity comes with the benefit of electron acceleration and velocity control[25–29]. In particular, the conveyor-mode shuttling approach in Si/SiGe combines this advantage with predictable spin coherence during

[1]JARA-FIT Institute for Quantum Information, Forschungszentrum Jülich GmbH and RWTH Aachen University, Aachen, Germany. [2]ARQUE Systems GmbH, Aachen, Germany. [3]Helmholtz Nano Facility (HNF), Forschungszentrum Jülich, Jülich, Germany. ✉e-mail: lars.schreiber@physik.rwth-aachen.de

shuttling and the requirement for just four input signals independent of the length of the shuttle device[11]. High-fidelity short-range conveyor-mode charge[30] and spin[31] shuttling have been demonstrated.

In this work, we all-electrically position and detect up to 34 electrons in a single-electron conveyor-mode QuBus in Si/SiGe. Despite its unprecedented length of 10 μm and more than 100 electrostatic gates, the QuBus can be controlled by only six input terminals with low voltage pulse complexity. We introduce a characterisation method we call shuttle tomography to benchmark the local shuttle fidelity of the QuBus using a single electron as a probe. By composing elementary pulses, we can control and detect any single electron pattern filling a series of 34 QDs. The conveyor-mode shuttle approach opens up new possibilities for probing local potential disorder in a quantum well, detecting single-electrons with high lateral resolution across a length of 10 μm and boosting multi-electron control for scalable spin qubit quantum computation.

## Results

### QuBus device and pulse segments

Our QuBus device consists of an undoped SiGe/Si/SiGe quantum well on top of which three electrically isolated metallic gate layers are fabricated by electron-beam lithography and metal lift-off (see "Methods" for details on the device fabrication). The 10 μm long grounded split-gate on the first layer defines a nominally depleted one-dimensional electron channel (1DEC) in the quantum well. More than 100 clavier gates, equally distributed among the second and third layer above the 1DEC enable the approximately uniform movement of single electrons along the $x$ direction (Fig. 1a) even in the presence of electrostatic disorder[11]. Notably, every fourth clavier gate is electrically connected to one of four gate sets $S_i$ ($i = 1 \ldots 4$). The connection scheme of four gate sets can be easily implemented as our clavier gates are distributed among two electrically isolated layers. At a fixed

distance between 1DEC and gate layers, using more than four gate sets would smooth the propagating potential towards a sinus function at all shuttle times, while with a cost of more complex gate design and gate connectivity. The simulated electrostatics of the conveyor-mode shuttling in our device is visualised in the Supplementary Video 1.

On-demand, a single electron can be loaded into the 1DEC from the left single-electron transistor (SET) formed by the gates TGL, LB1, LP and LB2. The plunger gate TLP of the leftmost quantum dot $QD_0$ controls the loading of exactly one electron from the SET to $QD_0$. The corresponding voltage pulses in gate space are indicated in Fig. 1b from the yellow dot to the blue square. This is followed by raising the tunnel barrier by gate TLB1 (pink triangle in Fig. 1b, c). We label the corresponding pulse segment as $P_1$. If the voltage $V_{TLP}$ applied to gate TLP remains low during the entire segment, no electron is loaded which we label as $P_0$ (Fig. 1c). Reversely, we can also use the SET current $I$ to detect either zero or one electron in $QD_0$ by the pulse segment $D_L$ (Fig. 1d). The detection pulse includes the unloading of the electron (see Supplementary Fig. 1 for details on the charge detection).

To shuttle the single electron in a moving QD, simple sinusoidal voltage pulses $V_{S_i}(t)$ are applied to the gate sets $S_i$:

$$V_{S_i}(t) = A_S \cos\left(2\pi f t - \frac{\pi(i+1)}{2}\right) + B_s + \Delta B_s \frac{1 + (-1)^i}{2}, \quad (1)$$

where the pulse amplitude $A_S$ sets the confinement strength of the propagating sinusoidal potential created in the 1DEC. $B_S$ and $\Delta B_S$ are constant offsets for accumulating charges in the conduction band in the 1DEC, accommodating different distances of the gate sets from the 1DEC. Both $B_S$ and $\Delta B_S$ are independent of the shuttle process and the electron position and thus remain constant for all pulse sequences. This significantly eases the operation of the conveyor-mode shuttle device. The shuttle velocity is given by $f\lambda = 14\ \mu m \cdot s^{-1}$, where the

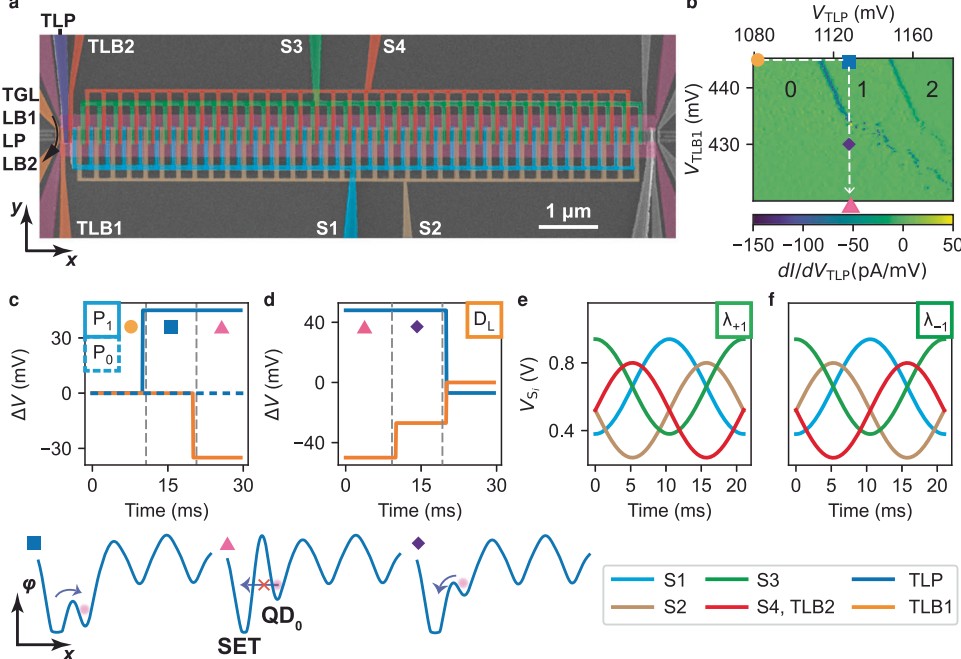

**Fig. 1 | QuBus device and pulse segments. a** False-coloured scanning electron micrograph of a top-view on a device nominally identical to the measured device. The gate labelled TGL overlaps with two barrier gates LB1 and LB2 and accumulates the electron reservoirs for the left SET. The gate labelled TLB2 is electrically connected to terminal $S_4$. **b** Charge stability diagram for controlling the $QD_0$ filling (numbers) by individually pulsed terminals TLP and TLB1. Symbols indicate positions in gates space consistent with (**c, d**). The line-cut of the electrostatic potential of the SET and $QD_0$ taken along $x$-direction are sketched for three positions

beneath (**c, d**). **c** Voltage pulses $\Delta V$ for loading one (solid line) and zero electrons (dashed line) labelled $P_1$ and $P_0$, respectively. **d** Voltage pulse $\Delta V$ of the segment labelled $D_L$ for detecting an electron in $QD_0$ by the left SET. After detection the electron is unloaded to the SET. **e, f** Pulse segments $\lambda_{+1}$ and $\lambda_{-1}$ for shuttling an electron by a distance $\lambda$ in positive (**e**) and negative (**f**) $x$-direction using all terminals $S_i$ (see legend). All unchanged voltages during pulse segments are not plotted in panels c to f. **c – f** share the same legend illustrated on the bottom right. $\Delta V$ are voltage pulses with respect to the zero-point marked by the yellow dot in (**b**).

frequency of the shuttle pulse is $f = 50$ Hz and $\lambda = 280$ nm is the lateral period of the potential in the 1DEC.

To transport all electrons in the 1DEC by a distance of $\lambda$ in the positive (negative) $x$-direction, the pulse segment $\lambda_{+1}$ ($\lambda_{-1}$) is employed (Fig. 1e, f). Note that all voltages applied to gates of the device return to their initial values at the end of each $\lambda_{\pm1}$ pulse segment. This implies that the correction of the SET's operating point for capacitive cross-coupling to the clavier gate sets $S_i$ is constant and thus simple. The right SET and the rightmost clavier gates are not used here and voltages are chosen to have an open 1DEC towards an energetically lower lying electron reservoir. Thus, the pulse sequences we developed could be applied to shuttle devices with just one SET detector. Specifically, the shuttle tomography presented below could be used to identify the position of a faulty break in a shuttle device. In total only six voltage pulses $V_{S_i}$, $V_{TLP}$ and $V_{TLB1}$ given by the elementary pulse segments $P_0$, $P_1$, $\lambda_{+1}$, $\lambda_{-1}$ and $D_L$ control the whole 10 µm long shuttle device, inside of which a total of 35 QDs (QD$_i$ with $i = 0 \ldots 34$) are formed along the 1DEC.

## Single-charge shuttle tomography

In order to discuss the composition and interplay of pulse segments during the operation of the QuBus, we choose the pulse sequence called shuttle tomography as a first example. The sequence is designed to measure the local shuttle fidelity $F_\lambda$, i.e. the shuttle success rate of $\lambda_{+1}$ for a specific position of the probe electron. Thus, we might identify local weak spots in the QuBus, although charge detection by the SET is limited to QD$_0$. The strategy is to shuttle a single electron from the left end of 1DEC further into the 1DEC by some short distance and then back to the detector. We repeat this experiment in order to record the shuttle fidelity and sequentially increase the shuttle distance until the electron is shuttled the full distance of 19 µm forth and back. The obtained data serves as a benchmark of the local shuttle fidelity in the QuBus. The corresponding pulse sequence is displayed in Fig. 2a (see Supplementary Fig. 2 for SET current traces). First, the depleted 1DEC is loaded with a single electron ($P_1$) which is then shuttled into the 1DEC for a distance of $n \cdot \lambda$ (by repeating $\lambda_{+1}$ $n$-times). Afterwards, the electron filling of the first $n + 4$ QDs is measured by consecutively detecting ($D_L$) ($n + 4$)-times (see Fig. 2a) and shuttling one period back towards the SET ($\lambda_{-1}$). First shuttle tests on this device revealed that measuring 4 additional QDs is sufficient to detect an erroneously shuttled electron. Finally, we apply a reference pulse by repeating the full pulse, but replace the $P_1$ segment by $P_0$. This shuttle pulse is repeated $N$-times.

As an instructive subset of such a measurement over $N = 40$ pulse repetitions with $n = 7$ and $A_S = 245$ mV, we observe the filling of each of the first 11 QDs as shown in Fig. 2b. During the majority of shuttle pulses the electron remains within QD$_7$, into which it was shuttled. This result indicates a well-operating QuBus. Sometimes the electron is detected in QD$_8$ and QD$_9$, thus the shuttle process failed during these repetitions. Via the reference pulse segment, we check whether electrons leak into the 1DEC. Since we never observe any electrons during the reference pulse across thousands of repetitions for all $n$, we conclude that there is no such leakage and the SET charge-detector does not faultily detect electrons in an empty QD.

The full observation of shuttle tomography with $A_S = 280$ mV, $N = 1000$ and $n = 1 \ldots 36$ shows that the single electron is nearly always detected in the expected QD$_n$, into which it has been loaded (Fig. 2c with more details in Supplementary Fig. 3). In addition, no electrons are observed for $n = 35, 36$. This is expected as the right end of the 1DEC is open and the 1DEC only contains QD$_0$ to QD$_{34}$. Hence, the electron is pushed out of the 1DEC through its right end for $n = 35, 36$.

We introduce the electron count $C_m^l$ to express the number of electrons detected in QD$_m$ summing over all $N$ pulse repetitions where $l$ is the expected filling of QD$_m$, which is 1 only for $n = m$ and 0

otherwise. Thus, the error count of each QD relative to its expectation is given by $\Delta C_m^l = C_m^l - l \cdot N$. The single-electron error count (Fig. 2d) reveals that in very few repetitions the electron was detected in a QD$_m$ with $m > n$ and almost never for $m < n$. In $\simeq 1\%$ of the repetitions the electrons seem to disappear ($\sum_{m,l} C_m^l < N$) (see Supplementary Fig. 3). Remarkably, some electrons are detected in QD$_{35}$ when loaded in QD$_{34}$, although QD$_{35}$ does not exist. Hence, delayed electrons got stuck during a $\lambda_{-1}$ pulse segment, instead of hopping over one QD during a $\lambda_{+1}$ pulse segment. This indicates a directionality of the shuttle error. In other words, if we shuttle an electron to QD$_n$ and detect an electron in e.g. QD$_{n+1}$ (Fig. 2d), the electron has never reached QD$_{n+1}$. We apparently detect it there, because a shuttle error (a delay) occurred at $\lambda_{-1}$, if shuttling back to the SET.

We define one shuttle pulse as successful, if three conditions are simultaneously fulfilled: (I) An electron is detected in the $n$-th QD, into which an electron has been loaded. (II) No electron is detected in all other QDs, which are detected during the sequence. (III) No electron is detected during the reference shuttle sequence in any QD. We count the number of successful shuttle pulses with the same $n$ and divide by the total number $N$ of pulse repetitions to get the charge shuttle-fidelity $F(n)$.

With $A_S = 280$ mV and $N = 1000$ for each of the $n = 1 \ldots 34$ covering a shuttle distance of $2n \cdot 280$ nm, we observe an average shuttle infidelity of $1 - F = (0.785 \pm 0.051)\%$ (Fig. 2e). This infidelity, however, also includes errors from loading and detection pulse segments. Remarkably, $F(n)$ is almost independent of the shuttle distance. Therefore, we split the observed infidelity into two error sources: first, the shuttle error $\varepsilon_\lambda$ occurring during each $\lambda_{\pm1}$-pulse, which is a shuttle dependent error that accumulates over the increment of shuttle periods. Second, an electron loading and detection (LD) error $\varepsilon_{LD}$, which is independent of shuttling and attributed to errors occurring during the $P_1$ (no electron initialised by error) and $D_L$ (no electron detected by error) pulse segments. We linearly fit $\ln(F) = A \cdot n + B$ where $A = 2\ln(F_\lambda) = 2\ln(1 - \varepsilon_\lambda)$, $B = \ln(F_{LD}) = \ln(1 - \varepsilon_{LD})$ and find the average shuttle fidelity per period $F_\lambda = (99.996 \pm 0.003)\%$ at $A_S = 280$ mV corresponding to a simulated orbital splitting of 4 meV between the ground and the first orbital excited state (see Supplementary Fig. 4). The LD error is $\varepsilon_{LD} = (0.7 \pm 0.1)\%$, thus the LD-corrected shuttle fidelity across the full channel and back is $\hat{F}(34) = (99.7 \pm 0.3)\%$ (total distance 19 µm, see 'Methods' for details on the estimations of errors). Presumably, larger $A_S$ could increase the shuttle fidelity as long as heating effects remain negligible.

Finally, we provoke shuttling errors by reducing $A_S$ and thus the confinement of the QDs in the shuttle potential. Note that the amplitude of the flush pulse is always constant at $A_S = 280$ mV. We observe that as we decrease $A_S$, the shuttle fidelity drops between the third and the fourth shuttle period and then remains constant (left insert in Fig. 2e). Thus, we attribute the decrease in $F$ to a local weak spot in the QuBus potential, likely due to static potential disorder. To confirm this hypothesis, we modify the shuttling tomography pulse sequence by tuning $A_S$ as a function of shuttle distance. Therefore, temporarily enhanced confinement is realised by keeping $A_S = 280$ mV during the fourth $\lambda_{+1}$ and the $(n - 3)$-rd $\lambda_{-1}$ pulse segment, thus at the position of the weak spot only. This demonstrates a tunable method to shuttle electrons over the QuBus with high $F$ at much lower $A_S$ applied during all other pulse segments $\lambda_{\pm1}$ (right inset of Fig. 2e). The observed cut-off amplitude at 100 mV matches well with simulations of semiconductor-oxide interface charge-defect induced potential disorder in the 1DEC[11]. The origin of the weak spot in the QuBus requires further investigation. Note that for the measurement of $F(n)$, we cannot fully exclude two errors appearing during shuttling which compensate each other. However, the observation that faulty shuttling behaviour occurs locally in the QuBus makes it probable that two such spots should be separately observed by the $n$-dependence of the shuttling tomography.

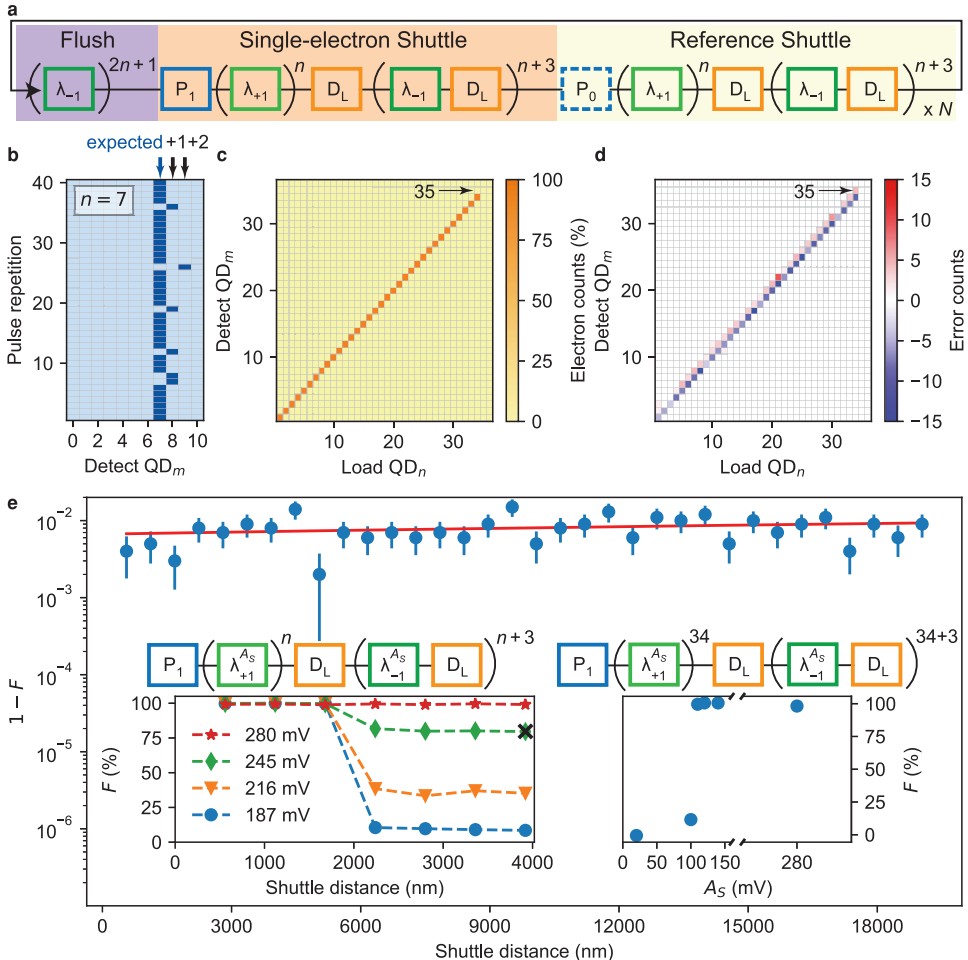

**Fig. 2 | Single-charge shuttle tomography. a** The pulse sequence of a shuttling tomography experiment consists of a flush pulse, a single-electron shuttle sequence and a reference shuttle pulse without an electron being loaded. The pulse is repeated $N$-times. **b** Digitised electron-detection map recorded during the single-electron shuttle pulse for $A_S = 245$ mV and $n = 7$. By single-shot charge-detection $(D_L)$ of the $QD_0$ to $QD_{10}$, we either find zero (light blue) or one (dark blue) electron for each of the 40 pulse cycles. The expected occurrence of an electron is indicated by the blue arrow and faulty locations by black arrows. The $F$ for this specific data set is marked by a cross in the left inset of (**e**). **c** Relative error counts i.e. fraction of electrons detected in $QD_m$, if one electron is loaded in $QD_n$ for $A_S = 280$ mV and $N = 1000$. For each column, we count all single-electron detection event from an electron-detection map as shown in (**b**). **d** Same as in (**c**) but the error counts are

plotted, i.e. the difference between expected and detected electron counts. The statistics shown in (**c**, **d**) are based on the same data set. False electron detection of $QD_{35}$ is marked by an arrow. **e** The shuttle infidelity $1 - F$ (blue dots with $1\sigma$ error bar) determined from the full shuttle-tomography sequence for $A_S = 280$ mV and $N = 1000$. The number of applied $\lambda_{\pm1}$-pulse segments is converted into total single-electron shuttle distance forwards and backwards. Red-line is fit to the data (see text). Left inset: Shuttle fidelity $F$ as a function of the total shuttle distance (forth and back) for various $A_S$ and $N = 1000$ (lines are guide-to-the-eyes). Right inset: Shuttle fidelity $F$ as a function of $A_S$ for maximum shuttle distance $n = 34$. Note that $A_s = 280$ mV is only used for the fourth $\lambda_{+1}$ and for the 31st $\lambda_{-1}$ shuttle pulses. The modified single electron shuttle pulses are sketched above the corresponding inserts. The applied flush and reference pulses are not shown for simplicity.

## Multi-electron operation

For the shuttle tomography, only exactly one electron was loaded into the QuBus at a time. The QuBus can also be operated with many electrons using the aforementioned elementary pulse segments. Each electron can be placed in any of the QDs between $QD_1$ and $QD_{34}$ in a controlled manner. Thus, we can create a pattern of electron fillings in a 34 QD register. The pulse sequence for loading and detecting an arbitrary electron pattern in the QuBus (Fig. 3a) is similar to the sequence employed during shuttle tomography. The repetition of $\lambda_{+1}$ segments is replaced by a series of single $\lambda_{+1}$ interleaved with $P_0$ and $P_1$ pulse segments. The latter determine the pattern filling the QDs. The key expectation is that any $\lambda_{\pm1}$ segment should move all electrons simultaneously by shifting the sinusoidal potential in the 1DEC.

Using $A_S = 280$ mV, we load one electron in each of the 34 QDs (1111...), every second QD (1010...) or a more complex periodic pattern (1100...). We repeat the pattern loading and detecting for $N = 1000$ times to gather statistics on the electron count in each QD

(Fig. 3b). We observe that the fraction of counted electrons in all QDs is very close to the expected filling pattern. Next, nine non-periodic patterns P1-P9, representing the lines of a binary image comprising $34 \times 9$ bits, are successfully loaded and detected as observed from the statistics of $N = 100$ pulse repetitions.

The dominant bluish colour in the error-count map for all patterns (Fig. 3c) reveals that the main error is the apparent loss of electrons. We assign this notion to the dominance of the loading and detection error $\varepsilon_{LD}$. It also explains why the pattern fidelity, which we define analogue to the shuttle-fidelity as the rate of successfully and exclusively placing and detecting electrons in all intended QDs, is lowest for the pattern with the highest electron count (111...). Blue/red dipoles in the error-count map indicate a shuttle error. As for shuttle tomography, we mainly observe individual electrons being misplaced by one QD to the right, provided this adjacent QD is nominally empty. This observation underlines the directional character of the shuttle error, which we already noted for the shuttle tomography.

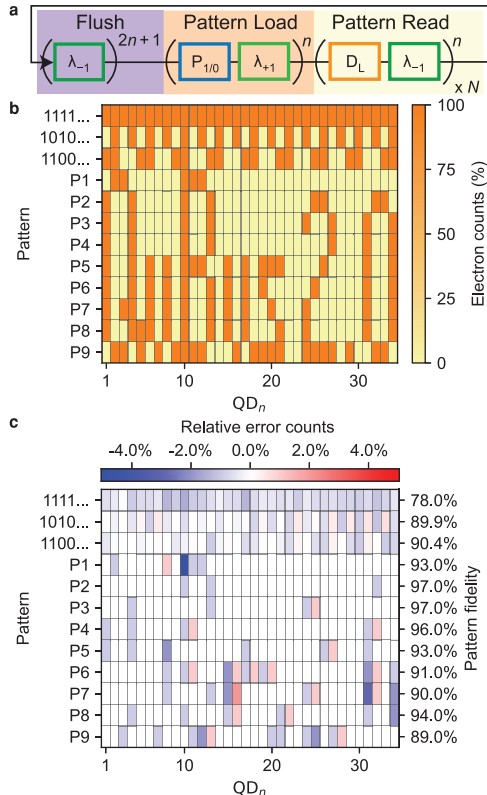

**Fig. 3 | Multi-electron operation. a** Pulse sequence composed of elementary pulse elements. **b** Detected number of electrons in $QD_n$ normalised by the total number of pulse repetitions $N$ for an arbitrary target filling pattern (rows with labels on the left). For $P_i$, the target pattern is non-periodic but can be identified from the data due to the low shuttle error and is therefore not explicitly given. **c** The relative error counts of each QD for shuttled patterns. The pattern fidelity, as defined in the text, is shown on the right for each individual row.

## Discussion

Our all-electrical Si/SiGe electron-shuttle device is successfully operated in conveyor mode. Only six input terminals control the more than 100 clavier gates of the 10 μm long device. Independent of its length, only four sinusoidal signals are required to operate the shuttle as well as two signals for loading and detecting electrons by a single-electron transistor. We introduce a method called shuttle tomography, which uses a single electron to probe the local shuttle fidelity and thus local imperfections in the confinement of the moving QD. We estimate the fidelity for shuttling one electron across the full length and back, thus a total distance of 19 μm, to be $\hat{F} = (99.7 \pm 0.3)\%$. Employing other pulse sequences composed of the five elementary pulses for our QuBus, we programmatically distribute and detect up to 34 electrons across the 34 QDs formed in the shuttle device. Any QD filling pattern can be initialised and we encode a digital image, the pixels of which are represented by single electrons.

The QDs can be interpreted as a 34 bit stack with a maximum of 34 electrons or as the initialisation procedure of a series of 34 QDs towards a quantum register for spin qubits. Preparing such patterns of electrons in a one-dimensional channel with low electrostatic disorder opens up possibilities to study the interplay of tunnel coupling and Coulomb interaction for a specific charge configuration. For example, as the amplitude of the periodic shuttle potential set by $A_S$ is lowered and tunnel coupling between neighbouring QDs increased, electrons become ordered by their Coulomb interaction. The reordered position of electrons might be observed by ramping up the periodic potential and readout the filling of each QD. Furthermore, our conveyor-mode QuBus device paves the way to scalable quantum computation, since it

is expected that the electron-spin evolution is deterministic during conveyor-mode shuttling at a velocity of approximately 8 m · s$^{-1}$ and spin-coherent shuttle fidelities of 99.9% are predicted[11]. Besides the coherent spin-shuttling demonstrated for shorter distance[31], our current work gives perspectives for shuttle fidelity of the QuBus across ≈ 10 μm required for a sparse and scalable quantum computation architecture[16,17]. Notably our QuBus is technologically compatible to industrial fabrication and Si/SiGe has been proven to be an ideal host-crystal for spin qubits. Spin-qubit connectivity across a distance of several micrometres could be a game changer for spin quantum computation.

## Methods

### The QuBus device

The undoped quantum well heterostructure is grown by chemical vapour deposition on a 200 mm silicon wafer and consists of a 7-nm tensile-strained silicon layer sandwiched between two relaxed layers of $Si_{0.70}Ge_{0.30}$. The upper barrier layer of $Si_{0.70}Ge_{0.30}$ has a nominal thickness of 30 nm and is capped by 2 nm of Si. Ohmic contacts to the quantum well are created by the selective phosphorus ion-implantation followed by a rapid thermal anneal at 700 °C for 30 s. The contacts are then metalized using optical lithography and metal lift-off. Three metallic gate layers including fan-out are fabricated via electron beam lithography and evaporation followed by metal lift-off. A scanning electron micrograph of a device nominally identical to the device measured in this work can be seen in Fig. 1a. The first gate layer is deposited directly onto the silicon capping layer, the native oxide layer of which was removed immediately before metal evaporation via HF etching. For this lowest layer 15 nm of palladium is used in order to fabricate a Schottky contact to the Si-cap. According to the electrostatic simulations[11], this results in a lower potential disorder in the quantum well. The later two gate layers are fabricated on 7 nm of atomic layer deposited $Al_2O_3$ and consist of 5 nm of titanium and 22/29 nm of platinum for the second and third layer, respectively.

The first fine gate layer defines both the SET plunger and barrier gates as well as the channel-confining split-gate. The split-gate constrains the 1DEC to a width below 200 nm. The second and third metal gate layers define the SET top gates as well as the clavier gates, which form the individual QDs in the 1DEC. These clavier gates have a width of 60 nm with a pitch of 70 nm. The designed distance between SET and $QD_0$ and thus the tunnel-coupling is based on ref. 32.

### Experimental setup

Experiments are conducted in an Oxford Triton 200 dilution refrigerator at ≃ 60 mK. Voltage pulses are generated by a Zurich Instruments HDAWG8 and superposed with DC voltages from a home-built DAC by a passive voltage adder at room-temperature. All signal lines are filtered by Pi-filters with a cut-off frequency of 1 kHz. No low-temperature filtering is used. The SET current is converted by the low-noise transimpedance amplifier SP983c from Basel Precision Instruments with a cut-off frequency of 3 kHz and digitised by an AlazarTech ATS9440 waveform digitiser. The composition of pulse sequences employs the open-source Python package qupulse[33].

### Error estimation

Here we discuss and estimate the error probability for manipulating the charge state during each elementary pulse segment $\varepsilon_i$ with $i = P_0, P_1, \lambda_{+1}, \lambda_{-1}, D_L$. First, we assume for simplicity that the average error for shuttling one electron by a distance of $\lambda$ is $\varepsilon_\lambda \approx \varepsilon_{\lambda_{+1}} \approx \varepsilon_{\lambda_{-1}}$ despite the experimentally observed small directionality. Since we never observe any electrons during the reference pulse across thousands of shuttle tomography repetitions for all $n$, we conclude that $\varepsilon_{P_0} \approx 0$ and that the detector does not faultily detect electrons in an empty QD. We combine the error from loading one electron $\varepsilon_{P_1}$ and missing an electron during detection $\varepsilon_{D_L}$ to be the loading and

detection error $\varepsilon_{LD}$ with $(1 - \varepsilon_{LD}) = (1 - \varepsilon_{P_I})(1 - \varepsilon_{D_L})$. The experimentally observed shuttle fidelity $F(n)$ during a shuttle tomography pulse sequence of shuttle distance $2n\lambda$ is composed of several elementary pulse segments:

$$F(n) = (1 - \varepsilon_{LD}) \cdot (1 - \varepsilon_\lambda)^{2n}. \qquad (2)$$

By linearly fitting $\ln(F(n))$, we find $\varepsilon_{LD} = (0.7 \pm 0.1)\%$ and the average shuttle fidelity per period $F_\lambda = 1 - \varepsilon_\lambda = (99.996 \pm 0.003)\%$. Thus, the expected LD-corrected shuttle fidelity $\hat{F}(n) = (1 - \varepsilon_\lambda)^{2n}$ for a total shuttle distance of $\approx 19\,\mu m$ is $\hat{F}(34) = (1 - \varepsilon_\lambda)^{68} = (99.7 \pm 0.3)\%$. This corresponds to the fidelity of shuttling a single electron across the full QuBus and back. Note that we distinguish $\varepsilon_\lambda$ from $\varepsilon_{LD}$ by increasing the number of $\lambda$ pulses.

## Data availability
The data generated in this study have been deposited in the Zenodo database (https://doi.org/10.5281/zenodo.8375442).

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

## Acknowledgements

The authors thank Łukasz Cywiński for helpful comments on the manuscript. This work was funded by the German Research Foundation (DFG) within the project 421769186 (SCHR 1404/5-1) and under Germany's Excellence Strategy - Cluster of Excellence Matter and Light for Quantum Computing (ML4Q) EXC 2004/1 - 390534769 and by the Federal Ministry of Education and Research under Contract No. FKZ: 13N14778. Project Si-QuBus received funding from the QuantERA ERA-NET Cofund in Quantum Technologies implemented within the European Union's Horizon 2020 Programme. The device fabrication has been done at HNF - Helmholtz Nano Facility, Research Centre Juelich GmbH[34].

## Author contributions

R.X. conducted the experiments with M.B., L.R.S. R.X. and M.B. analysed the data with L.R.S. S.H. provided measurement software. T.S prepared the passive voltage adder. M.B. adapted the developed process by R.X., J., S.T. and I.S for the device fabrication and fabricated the device. S.T. operated e-beam lithography. L.R.S. designed the device and supervised the experiment. L.R.S. and H.B. provided guidance to all authors. R.X., M.B. and L.R.S. wrote the manuscript, which was commented by all other authors.

## Funding

## Competing interests
Conveyor-mode shuttling is covered by a patent family (EP 4031486, US 2022/0293846 A1, CN114424346 A) by inventors L.R.S, H.B., I.S. and-, Künne. The patent application, co-owned by RWTH Aachen University

and the Forschungszentrum Jülich, is currently pending. L.R.S. and H.B. are founders and shareholders of ARQUE Systems GmbH. The remaining authors declare no competing interest.
