## [Peer Review File · Nature Communications]

REVIEWERS' COMMENTS

Reviewer #1 (Remarks to the Author):

The authors addressed my concerns, and I am happy to recommend this manuscript for publication in NC in its current version.

Reviewer #2 (Remarks to the Author):

Again, I think this is a very nice manuscript. It has also been very thoroughly reviewed! The authors have also done a thorough job of responding to the reviewer comments, and I am satisfied with the result. In my opinion, the manuscript can be published in Nature Communications in its current form.

Reviewer #3 (Remarks to the Author):

In this paper, the authors demonstrate a charge transfer device in a Si/SiGe quantum dot structure. Due to the interdigitated gate design, they demonstrate that electrons can be moved across long distances with only a small number of gates. The loading and detection is performed with a single electron transistor at one end of the device. The paper is of significant interest, and should be published in Nature Nano. A few detailed comments are listed below:

* typos: missing word in the first sentence of the 4th paragraph.

* These results show the fidelity of charge transfer. For this approach to really impact quantum computing, it is important to understand the fidelity of the electron spin as it is moved along the chain. Can the spin fidelity of this device be assessed?

Reviewer #4 (Remarks to the Author):

I think the authors have done a good job responding to the comments from the reviewers, and the concerns have been addressed. I recommend publication in Nature Communications.

We would like to thank the four reviewers for the time taken to assess our manuscript as well as for the knowledgeable review. We think that the manuscript considerably improved by questions and comments of the reviewers. Here out point-by-point reply to the comments of the 4 reviewers.

Response to Reviewer #1

The authors addressed my concerns, and I am happy to recommend this manuscript for publication in NC in its current version.

We thank the reviewer for recommending our manuscript for publication in Nature Communications.

Response to Reviewer #2

Again, I think this is a very nice manuscript. It has also been very thoroughly reviewed! The authors have also done a thorough job of responding to the reviewer comments, and I am satisfied with the result. In my opinion, the manuscript can be published in Nature Communications in its current form.

We thank the reviewer for recommending our manuscript for publication in Nature Communications.

Response to Reviewer #3

In this paper, the authors demonstrate a charge transfer device in a Si/SiGe quantum dot structure. Due to the interdigitated gate design, they demonstrate that electrons can be moved across long distances with only a small number of gates. The loading and detection is performed with a single electron transistor at one end of the device. The paper is of significant interest, and should be published in Nature Nano. A few detailed comments are listed below:

We thank the reviewer for recommending our manuscript for publication in Nature Nanotechnology.

1. typos: missing word in the first sentence of the 4th paragraph.

Unfortunately, we cannot find the missing word in the sentence: "In this work, we all-electrically position and detect up to 34 electrons in a single-electron conveyor-mode QuBus in Si/SiGe."

2. These results show the fidelity of charge transfer. For this approach to really impact quantum computing, it is important to understand the fidelity of the electron spin as it is moved along the chain. Can the spin fidelity of this device be assessed?

We fully agree with the reviewer that the fidelity for moving a spin is crucial for quantum computing. We addressed this issue in a separate publication, which was accepted for publication in Nature Communications. In this work, we were limited by the signal bandwidth of the setup and therefore could not shuttle as fast as demonstrated in the other publication.

Response to Reviewer #4

I think the authors have done a good job responding to the comments from the reviewers, and the concerns have been addressed. I recommend publication in Nature Communications.

We thank the reviewer for recommending our manuscript for publication in Nature Communications.